# Tryptophan Modulation in Cancer-Associated Cachexia Mouse Models

**DOI:** 10.3390/ijms241613005

**Published:** 2023-08-21

**Authors:** M. Teresa Agulló-Ortuño, Esther Mancebo, Montserrat Grau, Juan Antonio Núñez Sobrino, Luis Paz-Ares, José A. López-Martín, Marta Flández

**Affiliations:** 1Laboratory of Clinical and Translational Oncology, Instituto de Investigación Sanitaria Hospital 12 de Octubre (Imas12), Av. Córdoba s/n, 28041 Madrid, Spain; agullo@h12o.es (M.T.A.-O.); lpazaresr@seom.org (L.P.-A.); 2Lung Cancer Group, Clinical Research Program, Centro Nacional de Investigaciones Oncológicas (CNIO), 28029 Madrid, Spain; 3Biomedical Research Networking Centre on Oncology—CIBERONC, Instituto de Salud Carlos III, 28029 Madrid, Spain; 4Department of Nursing, Facultad de Fisioterapia y Enfermería, Universidad de Castilla La-Mancha (UCLM), 45071 Toledo, Spain; 5Department of Immunology, Hospital Universitario 12 de Octubre, Av. Córdoba s/n, 28041 Madrid, Spain; esther.mancebo@salud.madrid.org; 6Animal Facility, Instituto de Investigación Sanitaria Hospital 12 de Octubre (Imas12), Av. Córdoba s/n, 28041 Madrid, Spain; montsegrau@h12o.es; 7Medical Oncology Department, Hospital Universitario 12 de Octubre, Av. Córdoba s/n, 28041 Madrid, Spain; juananrayo@hotmail.com; 8Medicine Department, Facultad de Medicina y Cirugía, Universidad Complutense de Madrid (UCM), 28040 Madrid, Spain; 9Faculty of Experimental Sciences, Francisco de Vitoria University (UFV), 28223 Pozuelo de Alarcón, Spain

**Keywords:** cancer, cachexia, inflammation, tryptophan, mouse models, skeletal muscle, 1-MT

## Abstract

Cancer cachexia is a multifactorial syndrome that interferes with treatment and reduces the quality of life and survival of patients. Currently, there is no effective treatment or biomarkers, and pathophysiology is not clear. Our group reported alterations on tryptophan metabolites in cachectic patients, so we aim to investigate the role of tryptophan using two cancer-associated cachexia syngeneic murine models, melanoma B16F10, and pancreatic adenocarcinoma that is KPC-based. Injected mice showed signs of cancer-associated cachexia as reduction in body weight and raised spleen weight, MCP1, and carbonilated proteins in plasma. CRP and Myostatin also increased in B16F10 mice. Skeletal muscle showed a decrease in quadriceps weight and cross-sectional area (especially in B16F10). Higher expression of atrophy genes, mainly Atrogin1, was also observed. Plasmatic tryptophan levels in B16F10 tumor-bearing mice decreased even at early steps of tumorigenesis. In KPC-injected mice, tryptophan fluctuated but were also reduced and in cachectic patients were significantly lower. Treatment with 1-methyl-tryptophan, an inhibitor of tryptophan degradation, in the murine models resulted in the restoration of plasmatic tryptophan levels and an improvement on splenomegaly and carbonilated proteins levels, while changes in plasmatic inflammatory markers were mild. After the treatment, CCR2 expression in monocytes diminished and lymphocytes, Tregs, and CD8+, were activated (seen by increased in CD127 and CD25 expression, respectively). These immune cell changes pointed to an improvement in systemic inflammation. While treatment with 1-MT did not show benefits in terms of muscle wasting and atrophy in our experimental setting, muscle functionality was not affected and central nuclei fibers appeared, being a feature of regeneration. Therefore, tryptophan metabolism pathway is a promising target for inflammation modulation in cancer-associated cachexia.

## 1. Introduction

Cancer cachexia is a complex multifactorial syndrome characterized by involuntary and pathological weight loss, mainly due to skeletal muscle wasting. Patients suffer nutritional deterioration, weakening, fatigue, and edemas, reducing considerably patient’s quality of life and overall survival. Cancer-driven cachexia occurs almost in every type of cancer and affects nearly 50% of all cancer patients and it is a cause of death in 20–30% of them. Cachexia prevalence ranges from 70% in pancreatic cancer to 30% or less in other cancer types [1]. Although removal of primary tumors has been shown to result in the reversal of cachexia, it is important to note that the severity of cancer-associated cachexia does not correlate with the size of the tumor [2,3].

In addition of worsening the pathogenic process of cancer, skeletal muscle atrophy also interferes with therapeutic treatment, decreasing patient tolerance to several common cancer therapies. Moreover, several common chemotherapeutic agents are themselves known to worsen symptoms of muscle atrophy in this syndrome. Cachexia typically involves a combination of anorexia and altered metabolism, which together culminate in a hypercatabolic state in which the rate of muscle proteolysis overtakes protein synthesis [4].

To date, factors initiating cachexia are unknown. Cancer-associated cachexia was defined in 2011 by Fearon KC [5] and his definition was based in the loss of body weight and signs of systemic inflammation. There are no biomarkers for cancer cachexia as there is not enough clinical evidence to validate any of the proposed theories [2]. Furthermore, there are no effective treatments for cachexia and no official guidelines exist for management of this syndrome. Supplemental nutrition is used in most of the cases although patient responses have been mixed [4]. During the last years, multimodal approaches have been tested, including pharmacological therapy, to reduce systemic inflammation, counteracting hypercatabolic state and/or stimulating appetite, nutrition care, adapted physical activity, and psychosocial care [2,6,7,8]. However, clinical trials using these approaches have shown diverging and hopeless results on skeletal muscle alterations and other cachexia symptoms. Different factors may explain these results. First, the heterogeneity of the cohorts, as they include different types of tumors; second, the number of patients included is often small and; third, the advance stages of the patients recruited for these trials [2].

Previous studies from our group [9] concluded that plasma amino acids changes in cancer-derived cachexia patients had the potential to lead to the discovery of mechanisms involved in the pathogenesis of cachexia. Among them, the alteration in metabolites related to tryptophan degradation called our attention and it is in accordance with previous published data that associates tryptophane (Trp) metabolism with cancer development [10,11]. Moreover, low levels of tryptophan were correlated with skeletal muscle atrophy in patients with cancer and, therefore, tryptophan metabolism pathway has been proposed as a promising target for preventing and treating skeletal muscle atrophies [12,13].

Indoleamine 2,3-dioxygenase (IDO) is the first and rate-limiting enzyme in the kynurenine pathway responsible for the degradation of tryptophan and known to support immune-inhibitory effects. Many tumors overexpress IDO and it has largely been associated with poor prognosis. By driving IDO over-expression, tumors can create an immunosuppressive microenvironment that blocks the antitumor immune response. Timosenko et al. showed in 2016 that the first IDO inhibitor, 1-methyl-tryptophan (1-MT), combined with chemotherapy had a good efficacy in tumor rejection. Overall, the published results at systemic levels provides a strong rationale for therapeutic target of IDO [14,15]. In addition, immune checkpoint inhibitors lead to an increase in IDO expression, which promotes resistance to single agent therapy. To avoid this resistance, a combination of immune checkpoints inhibitors and IDO inhibitors showed promising results that are being considered in the clinic. However, the unexpected failure of epacadostat, an IDO1 selective inhibitor, in a phase III clinical trial challenged the efforts to inhibit this target as an anticancer compound [16].

In our study, we intended to determine Trp levels in tumor/cachexia development and to recapitulate human disease in two mice models, with B16F10 and KPC-injected cells. We then aimed to investigate the role of tryptophan in cancer cachexia development by inhibiting IDO with 1-MT to study the effects of Trp modulation on inflammation and skeletal muscle alteration in our murine models.

## 2. Results

### 2.1. Murine Models for the Study of Cancer-Associated Cachexia Recapitulate Human Disease

We studied two mouse models with tumors derived from different cell lines. We chose murine melanoma cells B16F10 and pancreatic adenocarcinoma cells KPC to inject subcutaneously in the flanks of C57BL/6J mice. Both cell lines derived from the same genetic background as the host, recapitulating whole immune and inflammatory responses. Figure 1A shows the experimental design for the in vivo experiments performed in this study. Cells were injected in the right flank and the tumor grew exponentially as shown in Figure 1B. Along the whole experiment, we refer to control mice as those injected only with matrigel, without tumoral cells. Tumor-bearing mice did not gain or lose weight, as expected in development of cachexia, while control mice gained body weight during the experiment (Figure 1C, measured at the time of sacrifice). The spleen of mice injected with tumor cells had a bigger size and weight (1.8–2 fold increased) compared with control mice (Figure 1D).

We measured two inflammation-related molecules to follow cachexia development in our models. First, Circulating Monocyte Chemoattractant protein-1 (MCP1/CCL2) has been suggested as biomarker of cancer cachexia from early stages of patients with pancreatic cancer syndrome [17,18]. Second, C-reactive protein (CRP) is a classic marker of systemic inflammation included in Fearon definition of cachexia. MCP1 plasma levels increased in tumor-bearing mice compared with control (2.40 ± 1.2 and 6.74 ± 3.1 fold increase in B16F10 and KPC models, respectively) (Figure 1E). Interestingly, at the time of sacrifice, CRP levels increased only when B16F10 cells originated the tumors, but not when it was originated by KPC (1.96 ± 0.51-fold increase in B16F10 injected mice versus control mice; 0.94 ± 0.09-fold change in KPC-injected mice) (Figure 1E). Levels of myostatin (MSTN or GDF8) are abundant in skeletal muscle, linked to muscle wasting [19] and elevated in cachectic cancer rodents [20]. As shown in Figure 1E, MSTN plasma levels increased slightly in B16F10 injected mice, 1.72 ± 0.86 times compared with control, and remain similar in KPC tumoral cells mice (increased 1.17 ± 0.23). Note that the kinetics of tumor growth and tumor volumes differed clearly in both models (Figure 1B), as occurs in human tumors originated in different tissues, and this can lead to the disparity in responses between them. Overall, from this part, splenomegaly and the increase in some plasma inflammatory markers point to a systemic inflammation in these mice during tumor growth.

The introduction of carbonyl groups into the amino acid side chains of proteins represents a marker of global protein oxidation and it is considered the major hallmark for oxidative damage proteins. These modifications are generated by different reactive oxygen species in blood, tissues, and cells [21]. Several studies have reported that oxidative stress is one of the most common mechanisms of cancer cachexia; it is linked to systemic inflammation and affects negatively immune functions [22]. Figure 1F shows how levels of carbonylated proteins increased upon tumor growth, more significantly in KPC model.

Regarding muscle alteration, a slight decreased was seen in quadriceps weight at the time of sacrifice (Figure 2A), but alterations in fibers cross sectional area (CSA mean, Figure 2B, and relative frequency, Figure 2C) were observed, although changes in KPC cells-injected mice were milder. In comparison with control mice, a reduction of 14.9% in B16F10 injected mice and 8.1% in KPC in average CSA was observed in the quadriceps (Figure 2B). When looking at the frequency distribution, median values ranged from 768.38 μm^2^ in control mice to 739.21 μm^2^, and 656.69 μm^2^ in KPC and B16F10 models, respectively (Figure 2C). In addition, transcript levels of genes linked to atrophy in skeletal muscle, such as *Atrogin1*, *Murf1*, and *Mstn*, increased in quadriceps of tumor-bearing mice (Figure 2D). According to previous results (Figure 2B,C), bigger and more significant changes were observed in B16F10 carrying mice; *Atrogin1* was approximately 2.85 higher, *Murf1* 2.05 times, and *Mstn* augmented up to 867.9 times compared with control mice levels. In the case of KPC mice, *Atrogin1* was 1.47 higher, *Murf1* 1.70 times, and *Mstn* augmented to 1.40 compared with levels in non-tumor-bearing mice. Inguinal white adipose tissue decreased with the injection of both tumor models (Figure 2E).

Weight lost or no gains, detection of systemic inflammation (splenomegaly and plasma markers), elevated oxidative stress (measured by plasma carbonylated proteins), and skeletal muscle alterations are all signs of cachexia. Taken together, our analysis suggested that B16F10 and KPC models recapitulate some clinical features of cancer-induced cachexia in human patients.

### 2.2. Plasma of Cachectic Cancer Patients and Tumor-Bearing Mice Showed Reduced Tryptophan Levels

We performed specific ELISA assays to confirm previous metabolomics results showing decreased levels of tryptophan in the plasma of cachectic cancer patients [9]. As shown in Figure 3A, tryptophan levels significantly decreased in patients with cachexia compared with cancer patients with no signs of weight loss (44.15 ± 2.15 μM versus 68.69 ± 3.48 μM).

In accordance with patients’ results, we found that plasma tryptophan levels in the mice used in this study were significantly reduced, as seen in Figure 3B,C. We measured plasma tryptophan levels at different time points in each model following the differences in tumor growth observed. Initial assays to set up our experiment showed that tryptophan levels were reduced as soon as 11 days after B16F10 melanoma cells inoculation (Figure 3B) and revealed oscillating levels through time in KPC-injected cells mice (Figure 3C). These results demonstrated tryptophan fluctuation during tumor development in these models as seen in humans. Therefore, it can be useful to test pharmacological interventions targeting tryptophan metabolism. 

### 2.3. 1-Methyl-Tryptophan Treatment Modifies Tryptophan Metabolism and Ameliorates Inflammation Signs in Tumor-Bearing Mice

To determine the effect of restoring Tryp levels in the plasma of tumor-bearing mice, we administrated the IDO inhibitor 1-MT, previously proven to have no toxicity [23,24] and we studied signs of cachexia and inflammation. Oral administration of 1-MT in drinking water started 13 days after the injection of B16F10 cells (Figure 4A), while in KPC was administrated from day 18 post-injection (Figure 4B). Timings were chosen according to the differences in tumor kinetic in the models used. A group of tumor-bearing mice was used as control by administration of alkaline drinking water, which is the vehicle used for 1-MT administration.

Plasma tryptophan levels increased significantly after the treatment in both models, as seen in Figure 4A,B, confirming the activity of the drug in our experiments. Tumor size was slightly smaller in those mice treated with 1-MT but did not reach significance (Figure 4C and Appendix A).

Tumor development in our models leads to an increase in spleen size most likely due to systemic inflammation (Figure 1D). Reduced degradation of tryptophan by addition of 1-MT resulted in a partial recovery of spleen weight (Figure 5A). In mice carrying B16F10 cells, plasma levels of MCP1, CRP, and MSTN decreased to some extent (Figure 5B). However, no changes were observed in mice injected with KPC cells, although MCP1 increased slightly in response to 1-MT (Figure 5C). As mentioned before, in KPC-injected cells mice, MCP1 increased notably and significantly during tumor growth while CRP and myostatin did not change significantly.

Measurement of plasmatic protein carbonyls revealed an almost complete reversion to control values after 1-MT treatment in both models (Figure 5D). As observed before, the KPC model presented more clear and significant changes in carbonylated proteins.

The decrease in spleen size and the almost completed recovery of plasma carbonylated protein levels, close to control values, pointed to a partial reduction in the systemic inflammation in tumor-bearing mice that had been treated with 1-MT.

Flow cytometry analysis of peripheral blood cells was performed to follow immune cell population responses in our experiments. Treatment with 1-MT did not result in clear changes in the number of monocytes identified as CD11b^+^, either classical inflammatory-Ly6C^high^ or non-classical/resident-Ly6C^low^ types. Interestingly, C-C Motif Chemokine Receptor 2 (CCR2) expression decreased after inhibition of Trp metabolism in both B16F10 and KPC models (Figure 6A and Figure 6B, respectively). The number of lymphocytes identified by CD3 expression and Tregs subtype (CD3^+^, CD4^+^, Foxp3^+^, and CD25^+^) did not change under inhibition of Trp metabolism. However, in B16F10 mice, CD127 expression increased significantly in the Tregs cells after treatment (Figure 6C), and remained unchanged in KPC mice. However, in KPC tumor-bearing mice, CD25 expression was increased in CD3^+^ CD8^+^ cells as shown in Figure 6D, whereas no changes were detected in B16F10.

### 2.4. Muscle Deteriorates with 1-MT Treatment

To elucidate how 1-MT treatment affects muscle wasting during tumor development, analysis of quadriceps was performed. Surprisingly, we found a decrease in quadriceps weight in 1-MT-treated mice compared with non-treated, referred to total body weight, at the time of sacrifice (Figure 7A). Histological analysis of cross-sectional area on H&E-stained sections from quadriceps showed a decrease in the mean of fiber areas in both mice models (Figure 7B). In B16F10-injected mice, the average cross-sectional area was 654.6 ± 92.9 μm^2^ in non-treated tumor-bearing mice and 626.2 ± 87.7 μm^2^ in mice treated with 1-MT, representing a reduction of 14.9% and 18.5% compared with control mice (768.9 ± 77.9 μm^2^), respectively. In mice inoculated with KPC cells, the average fiber area in tumor-bearing non-treated mice was 706.4 ± 33.5 μm^2^ and 627.2 ± 86.8 μm^2^ in tumor 1-MT-treated mice. These results represent a reduction compared with controls (768.9 ± 77.9 μm^2^) of 8.1% in tumor non-treated mice and 18.4% in tumor 1-MT-treated mice. A representative picture of the H&E staining of quadriceps analyzed is shown in Appendix A.

Moreover, frequency distribution demonstrated a clear shift towards smaller areas in the fibers after 1-MT treatment compared with both control and tumor non-treated mice (Figure 7C). Median of fiber areas were 656.7 μm^2^ in B16F10 tumor-bearing mice moving towards 619.3 μm^2^ when treated with 1-MT. In accordance with the weight of the muscles at the time of sacrifice (Figure 7A), alteration was more drastic in KPC-injected mice with a median fiber area of 739.2 μm^2^ in non-treated mice and 651.1 μm^2^ in 1-MT-treated mice.

To further explore muscle alteration, we examined transcript levels of the atrophy-related genes studied previously in our study. Results confirmed a worsening of atrophy markers after 1-MT treatment in B16F10 injected mice while not much effect was observed in KPC-treated mice. As shown in Figure 7D, *Atrogene1* and *Murf1* mRNA levels approximately increased 3-fold and *Myostatin* showed a 2-fold increase compared with non-treated mice in the B16F10 model. In KPC-injected mice, atrophy-related genes did not show significant changes (Figure 7E). Note that transcript levels of these genes changed only slightly under tumor growth in KPC-injected mice. Following this tendency, inguinal fat decreased after 1-MT treatment in B16F10 mice but recovered partially in KPC-injected animals (shown in Appendix A). Although parameters measured suggested a worsening of skeletal muscle after 1-MT treatment in our experimental setting, no change in grip the strength test was observed and an increase in the number of fibers with central nuclei was observed, especially in B16F10 injected cells mice (Figure 7F). This phenomenon is known to be part of the regeneration process after injury in skeletal muscle. Altogether, muscle analysis after 1-MT treatment pointed to a worsening of muscle (mass, CSA, and atrophy markers) while muscle functionality is not affected and central nucleation suggests an active regeneration in these muscles.

## 3. Discussion

We found that B16F10 and KPC models recapitulate the main signs of cancer cachexia observed in humans in terms of weight changes, splenomegaly, increase in plasmatic MCP1 and levels of carbonylated proteins, and reduction in tryptophan in plasma and skeletal muscle alterations. Our data show that inhibition of Trp degradation during tumor development ameliorates inflammation while no benefit in skeletal muscle was detected in these mice.

Preclinical models reproducing closely the complexity of cancer cachexia are fundamental for the understanding of this syndrome. One of the most used in this context are the syngeneic mice [20], as used in this study. In addition, we consider that it is important to include different models as cachexia is found in a variety of tumor types in humans. Our mouse models, with a fast-growing cancer cell (B16F10) and a slower growing cancer cell (KPC) represent different origins and different stages of cachexia and may recapitulate different cachexia stages. We observed some differences between the two tumor models used for this study. Body weight loss, muscle alteration, and high CRP/Mst levels were more prominent in B16F10 tumor-bearing mice. On the contrary, MCP1 levels and the changes in carbonylated proteins detected in plasma were significantly more elevated in KPC tumor-bearing mice. We might find three possible explanations to the differences found in B16F10 and KPC-injected mice. First, note that the kinetics of tumor growth and final tumor volumes differed considerably in both models (Figure 1B). Each model is probably at a different stage at the time of sacrifice in terms of tumor development and, therefore, in inflammation and muscle alteration. Second, some of the well-accepted markers as systemic inflammation (e.g., CRP) are not consistently increased in cancer cachexia populations (reviewed in [2]) so, in a similar way, it may result in differences in the mouse models used for this study. Third, previously published data described different peaks of systemic inflammation. For example, Chiarella, P. et al. [25] reported in a fibrosarcoma induced murine cancer model two peaks of inflammation. An initial one detected very early, because of tumor cells inoculation, and a second peak coincidental with tumor growth and proportional to tumor size. In addition, two studies described that MCP1 plasma levels are higher in earlier stages of cachexia than later in the more advanced syndrome [17,18]. We also observed that myostatin levels in plasma did not correlate with their RNA expression levels (*MSTN* RNA level measured by qPCR) in muscle after 1-MT treatment, especially in B16F10 tumor-bearing mice (Figure 5B and Figure 7D). Also, atrophy regulators frequently show that protein levels may not necessarily parallel transcript levels due to increased protein turnover and decreased transcriptional rates in this context [26].

Considering our previous data [9] and eager to gain knowledge on pathogenic mechanisms in the development of cachexia, we explored Trp modulation in the context of cachexia. We also detected in our study that Trp levels decreased in both human and mouse plasmas during tumor development, reflecting that B16F10 and KPC cell injection might be used as models to study the role of tryptophan in cancer-associated cachexia. We then confirmed that treatment with the inhibitor of IDO 1-MT resulted in a recovery of the levels of plasmatic Trp in both models (Figure 4A,B) with a mild effect in tumor growth, an improvement in splenomegaly, a restoration in the levels of carbonylated proteins, and a decrease in CCR2 levels (receptor of MCP1), all of them linked to inflammation. However, after the treatment, only slight or no changes, were detected in the inflammatory markers measured in plasma. Our results go in line with the fact that kynurenines (metabolites derived from Trp degradation via IDO) have a close relationship with inflammatory signals and, therefore, can influence cancer progression [27]. In fact, an increased kynurenine to tryptophan ratio in blood is predictive for a worse outcome in cancer patients [11]. In terms of muscle atrophy, Ninomiya [13] demonstrated a positive relation between serum Trp levels and volume of skeletal muscles in patients that suffer diffuse large B-cell lymphoma. In the same study, mice fed with a tryptophan-deficient diet presented lower *tibialis anterior* fiber diameters than mice fed with a standard diet. In contrast to these data, inhibition of degradation of Trp in our experiments seems to exacerbate muscle atrophy in B16F10 and KPC models, according to muscle mass and CSA (Figure 7A–C). However, muscle functionality is not affected by these changes and the number of central nuclei fibers increased in treated mice. This last parameter appears in muscle regeneration after injury. It may occur that the improvement in systemic inflammation is not enough, in both time and strength, to result in a recovery of the muscle atrophy observed in these mice. Also, cachexia-associated muscle injury may be in part independent of the inflammatory kinetics and/or skeletal muscle mass might require longer time to benefit from this treatment.

We also consider that the muscle alteration may be an acute effect that will recover with time. This is supported by the fact that we did not find reports in the literature showing any secondary effect derived from skeletal muscle alterations. No weight loss or physical impairment after 1-MT treatment in mice have been described using longer experimental treatments than the ones used in our study [28,29]. Moreover, 1-MT treatment improved signs of stress in mice, reducing the loss of body weight and behavioral alterations after combined acoustic and restraint stress [30]. In addition, inactivity induces ROS production in skeletal muscle, which contributes to muscle atrophy. Paradoxically, evidence suggest that intracellular ROS production is required for normal remodeling that occurs in skeletal muscle [31,32]. Kynurenine derivatives are free radical generators, so kynurenine pathway contributes to increased oxidative stress [33]. It may happen that the worsening observed is the previous stage to the recovery of the skeletal muscle in our experiments. The observed reduction in carbonylated proteins indicates an improvement in oxidative stress in the plasma of mice after treatment (Figure 5D) that may lead to muscle recovery in a longer experimental setting. One of the limitations in our study is that the window for treatment may be too short to see a benefit from 1-MT.

CCR2 is the receptor of CCL2/MCP1, and their signaling determine the frequency of inflammatory monocytes in the circulation [34]. Increased levels of CCR2 and its ligand CCL2 promote tumor growth, are associated with inflammation, advanced cancer, and predicts prognosis and recurrence due to their role in the tumor microenvironment [35,36]. Also, this cytokine has been proposed as a potential biomarker of cancer-cachexia [17]. Studies in a pancreatic cancer model described that in the absence of CC-chemokine receptor 2 (CCR2) mice exhibit reduced anorexia and muscle atrophy. Moreover, pharmacological inhibition of CCR2 signaling decreased cachexia [37]. Interestingly, CCR2 expression decreased after inhibition of Trp metabolism in both B16F10 and KPC models (Figure 6A and Figure 6B, respectively) which may benefit inflammation in our mice models.

Regarding lymphocytes, the number of T lymphocytes (CD3^+^) and Tregs subtype (CD3^+^, CD4^+^, Foxp3^+^, and CD25^+^) did not change under inhibition of Trp metabolism. At the same time, we found that CD127 expression increased significantly in Tregs cells after treatment in B16F10 mice (Figure 6C) in line with the anti-inflammatory effect observed. CD127 is the α-chain of the IL-7 receptor and low levels of this molecule are correlated with suppressive activity of Treg cells [38]. CD127 expression inversely correlates with FoxP3 and suppressive function of human CD4^+^ Treg cells [39]. The decrease of Treg immunosuppressive activity may allow the activation of the specific antitumor adaptive cellular immune response and control the excessive nonspecific inflammatory response. In addition, in KPC tumor-bearing mice, CD25 expression was increased in CD8^+^ T cells as shown in Figure 6D. Upon activation, CD8^+^ T cells upregulate CD25 (α-chain of the IL-2 receptor) and become highly sensitive to IL-2 [40]. This result may indicate the recovery of the cellular adaptive immune response that may contribute to the antitumor fight. All together, these changes in immune population correlate with the improvement on inflammation observed in our mice after 1-MT administration.

Syngeneic models are useful because they are carried in immunocompetent mice, and tumor growth is fast, reproducible, and easy to manage. On the other hand, the tumor growth rate is a limitation (as in B16F10 model used in this study). Derived from this, the pharmacological window is too short, so we probably do not follow similar disease progression observed in humans. In addition, tumors are ectopic, which does not accurately reflect what is seen in patients.

In summary, our murine models recapitulated some human cancer-associated cachexia signs. By injection of melanoma and pancreatic cancer cells, we observed systemic inflammation and muscle atrophy during tumor growth in these mice. Treatment with 1-MT, an inhibitor of IDO, in these models decreased Trp degradation and resulted in an improvement in inflammatory signs linked to tumor development. While skeletal muscle did not show benefits in terms of wasting and atrophy, functionality was not affected and central nuclei fibers appeared, a feature of regeneration. Altogether, we believe that the tryptophan metabolism pathway is a promising target for cancer-associated cachexia due to its role in inflammation.

## 4. Materials and Methods

### 4.1. Patient Cohorts and Samples

This study was approved by the Ethics Committee of the “12 de Octubre” University Hospital (Ref. 10/294), and it was performed in accordance with the Declaration of Helsinki International Conference on Harmonization and Good Clinical Practice. Written informed consent was provided by all participating patients. Human samples used in this study were collected previously for the study published in [9].

### 4.2. Cell Culture

Melanoma (B16F10, from ATCC) and pancreatic adenocarcinoma (KPC: K-ras^LSL.G12D/+^; Trp53^R172H/+^; Pdx1-Cre, kindly provided by Dr. Bruno Sainz´s lab) cells were maintained in RPMI 1640 medium (Gibco, Waltham, MA, USA) supplemented with 10% fetal bovine serum (HyClone, Logan, UT, USA) and 1% antibiotic (Lonza, Basel, Switzerland) at 37 °C and 5% of CO_2_ and were periodically checked for contaminations.

### 4.3. Animal Experiments

For the animal studies, the Bioethical Committee of the Universidad Autónoma de Madrid and the competent authority approved the experimental protocol (Ref PROEX 218/19). All animal manipulations were made in accordance with the European Union guidelines. Male C57BL/6J mice (Charles River RMS Spain, Sant Cugat del Vallès, Spain), 6–8 weeks old were maintained on regular dark-light cycle, with free access to food and water during the whole experimental period. After 2 weeks of local animal care facility adaptation, animals were randomized into control-inoculated mice and tumor-bearing mice. Experimental groups were established with 9 animals.

Tumor cells inoculation: Cells were removed from culture flasks by adding 0.05% of trypsin solution (Gibco), centrifuged, and re-suspended in sterile PBS. Cell viability was determined by trypan blue exclusion. Finally, C57BL/6J mice were subcutaneously injected with a solution of 5 × 10^4^ B16F10 cells or 2.5 × 10^5^ KPC cells in a final volume of 100 µL (50 µL of cell solution mixed with 50 µL of Matrigel Corning), into the right flank. Control mice were inoculated with 50 µL of PBS mixed with 50 µL of Matrigel. At the indicated timings, blood was withdrawn from the jugular vein from anaesthetized mice (using sevoflurane) and collected in heparinized tubes, and then centrifuged (1500× *g*, 10 min 4 °C) to obtain plasma that was kept at –80 °C until used.

Tumor assessment: Tumor growth was monitored by a digital caliper three times. Tumor volume was calculated using a standard solid tumor formula V = 1/2 × (D × d2); where V is volume, D is higher diameter, and d is lower diameter. The same examiner performed all measures to minimize bias. Animals were monitored at least three times a week for body weight, tumor dimensions, and health condition, and euthanized 21 (B16F10 inoculated mice) or 39 (KPC inoculated mice) days after tumor cells inoculation. On the day of sacrifice, blood was withdrawn from anaesthetized mice (using sevoflurane) by cardiac puncture and collected in heparinized tubes, then centrifuged (1500× *g*, 10 min, 4 °C) to obtain plasma. Quadriceps were excised and, rapidly, snap frozen in liquid nitrogen. Spleen, liver, and tumors were collected and weighted right after.

### 4.4. 1-Methyl-Tryptophan Administration

The mice were given 5 mg/mL of 1-L-MT (Sigma-Aldrich, St. Louis, MO, USA) in their drinking water (alkaline water, pH 10.0, supplemented with 50 mg/l of aspartame) or just water with aspartame (control group) for the time indicated. The average amount of water drank was similar during the experiment in both groups (4–5 mL/d).

### 4.5. Enzyme-Linked Immunosorbent Assay (ELISA)

All ELISA assays were performed in plasma according to manufacturer´s instructions. Tryptophan (from ImmuSmol, Bordeaux, France); C-Reactive Protein (CRP), and Monocyte Chemoattractant Protein 1 (MCP1) (from Elabscience, Wuhan, China); Myostatin (from Cloud-Clone Corp., Katy, TX, USA).

### 4.6. Oxidative Stress Assays

Protein carbonyl groups were assayed in plasma using 2,4-dinitrophenylhydrazine adapted for a microplate reader [41] and expressed as nmol/mg total protein.

### 4.7. Reverse Transcriptase Quantitative PCR

For RNA extraction, tissues were first crushed in a pestle with liquid nitrogen. Total RNA was extracted using RNeasy Fibrous Tissue Minikit (Quiagen, Hilden, Germany) for skeletal muscle and RNeasy Mini Kit (Quiagen) for the rest of the tissues, following the manufacturer’s instructions. cDNA was reverse transcribed (NZY First-Strand cDNA Synthesis Kit, Nzytech, Lisboa, Portugal) and 50 ng RNA equivalent was used for PCR with specific primers (Table 1) in the presence of SYBR Green (NZYSpeedy qPCR Green Master Mix, Nzytech) using the 7500 Fast Real Time PCR System (Applied Biosystems, Waltham, MA, USA). All analyses were performed in duplicate. A melting curve analysis was performed for each reaction to control product quality and specificity. The expression levels of *Atrogin1*, *Murf1*, and *Myostatin* transcripts were normalized to individual hypoxanthine-guanine phosphoribosyltransferase (HPRT) expression using the ΔΔCt method.

### 4.8. Histology

Skeletal muscle was sectioned frozen fresh at 8 μm. H&E staining was performed using a standard protocol. Histological images were acquired with a Zeiss Axioplan2 microscope. Cross sectional area (CSA) of individual fibers was measured using ImageJ 1.53q software on 20× images. At least five randomly selected images were measured per muscle.

### 4.9. Flow Cytometry

For monocytes studies, whole blood was used for flow cytometry analysis. Briefly, 50 μls of fresh collected blood per mouse were transferred into a cytometry tube, antibodies were directly added to the tubes and incubated for 20 min at room temperature in the dark. A total of 1 mL of BD FACS Lysing Solution was added, mixed and then allowed 10 min to stand at room temperature in the dark. A total of 1 mL of PBS was added, centrifuged for 5 min at 1400 rpm, then the supernatant was discarded and the wash was repeated with PBS. We resuspended cells in 300 μls of PBS and which were then passed through the cytometer. For lymphocytes studies, mononuclear cells were isolated using a standard Ficoll protocol (250,000 cells/100 uls were used per staining). All antibodies, except for anti-Foxp3, were directly added to the tubes and incubated for 20 min at room temperature in the dark. After washing (add 1 mL PBS, centrifuge 5 min at 1400 rpm and discard supernatant), the manufacturer´s instructions were followed (Kit Foxp3 Staining buffer set, eBioscience) for permeabilization, incubation with the anti-FoxP3 antibody, and fixation of the cells. Antibodies used are listed in Table 2 and were used according to manufacturer´s instructions. Flow cytometry was performed using FACS Canto II (BD Biociences, Franklin Lakes, NJ, USA) y Navios (Beckman Coulter, Brea, CA, USA) and analyzed with FlowJo VX.

### 4.10. Statistical Analyses

The results are expressed as mean ± SD. Differences between groups were analyzed using the unpaired two-tailed Mann–Whitney test or unpaired two tailed *t*-test of the Graph Pad Prism program. Statistical significance was accepted at a *p* value < 0.05.

## Figures and Tables

**Figure 1 ijms-24-13005-f001:**
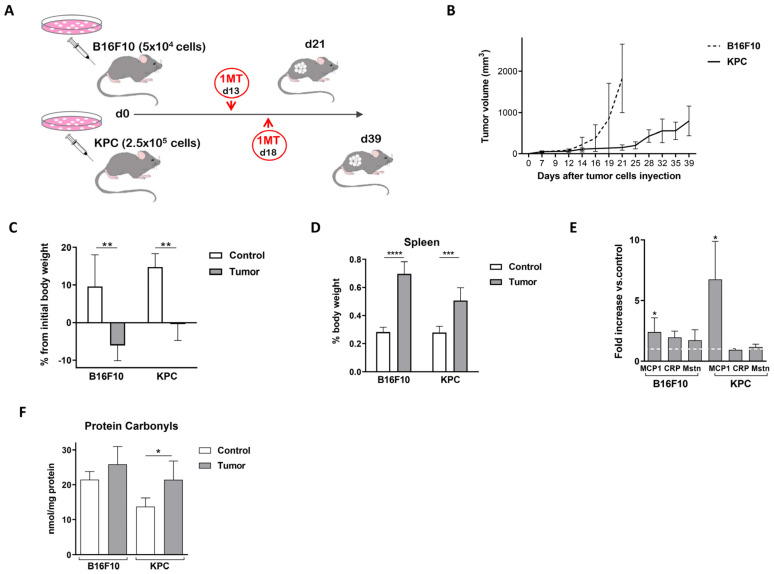
Murine models for the study of cancer-associated cachexia recapitulate human disease. (**A**) Experimental design for the in vivo experiments. (**B**) Tumor growth along the experiments. (**C**) Changes in body weight observed since injection of the tumor cells. (**D**) Spleen weight at the time of sacrifice referred to total body weight. (**E**) MCP1, CRP, and Mstn levels measured by ELISA in plasma and shown as fold increase versus non-tumor plasmatic samples. (**F**) Carbonylated protein levels measured in plasma. Significant differences between groups are indicated by * (*p* < 0.05), ** (*p* < 0.005), *** (*p* < 0.0005) and **** (*p* < 0.0001).

**Figure 2 ijms-24-13005-f002:**
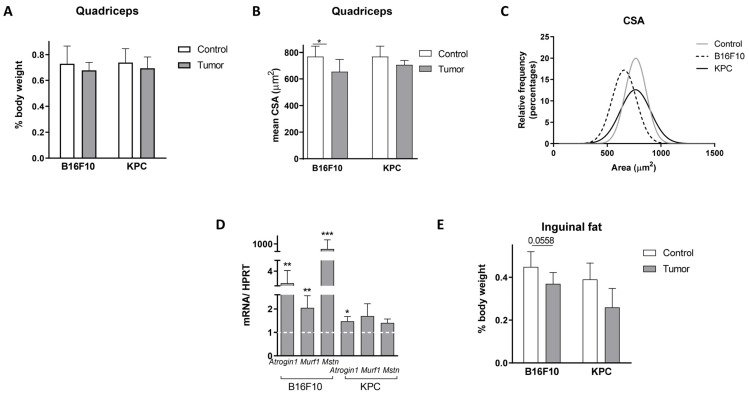
Muscle and inguinal fat alterations during cachexia development in B16F10 and KPC mouse models. (**A**) Quadriceps weight at the time of sacrifice referred to total body weight. (**B**) Quadriceps mean cross sectional measured in H&E sections. (**C**) Quadriceps cross sectional area measured in H&E sections and represented as relative frequency. (**D**) *Atrogin1*, *Murf1*, and *Mstn* RNA expression normalized by HPRT by qPCR. (**E**) Inguinal fat weight at the time of sacrifice referred to total body weight. Significant differences between groups are indicated by * (*p* < 0.05), ** (*p* < 0.005) and *** (*p* < 0.0005).

**Figure 3 ijms-24-13005-f003:**
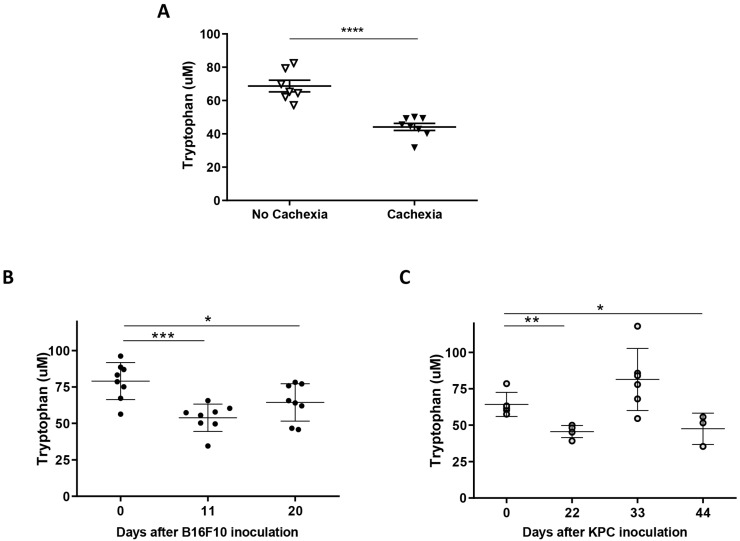
Plasma of cachectic cancer patients and tumor-bearing mice showed reduced tryptophan levels. Tryptophan levels measured by ELISA in plasma of cachectic and no cachectic cancer patients (**A**), B16F10 (**B**), and KPC (**C**) murine models during tumor development. Significant differences between groups are indicated by * (*p* < 0.05), ** (*p* < 0.005), *** (*p* < 0.0005), and **** (*p* < 0.0001).

**Figure 4 ijms-24-13005-f004:**
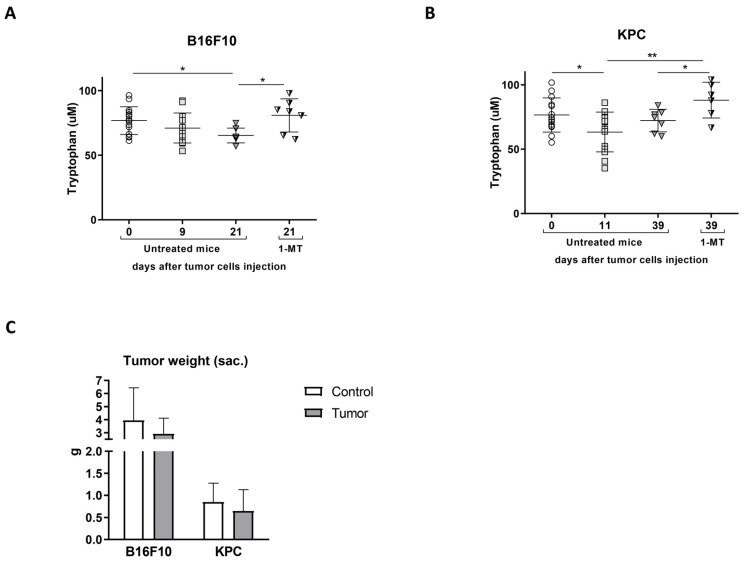
1-MT modifies tryptophan metabolism without clear effect in body weight and tumor growth. Tryptophan levels measured by ELISA in plasma of (**A**) B16F10 (**B**) and KPC comparing untreated and 1-MT-treated mice (different symbols represent different time points indicated in y axis). (**C**) Tumor weight at the time of sacrifice. Significant differences between groups are indicated by * (*p* < 0.05) and ** (*p* < 0.005).

**Figure 5 ijms-24-13005-f005:**
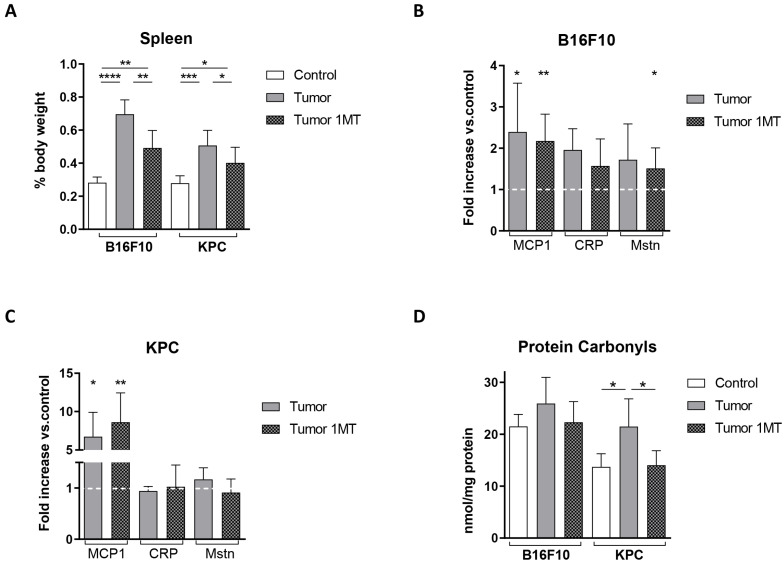
1-MT ameliorates inflammation signs in tumor-bearing mice. (**A**) Spleen weight at the time of sacrifice referred to total body weight. MCP1, CRP, and Mstn levels measured by ELISA in plasma (shown as fold increase versus non-tumor plasmatic samples) in (**B**) B16F10 and (**C**) KPC mouse models. (**D**) Plasmatic levels of carbonylated proteins in plasma of the same mice. Significant differences between groups are indicated by * (*p* < 0.05), ** (*p* < 0.005), *** (*p* < 0.0005), and **** (*p* < 0.0001).

**Figure 6 ijms-24-13005-f006:**
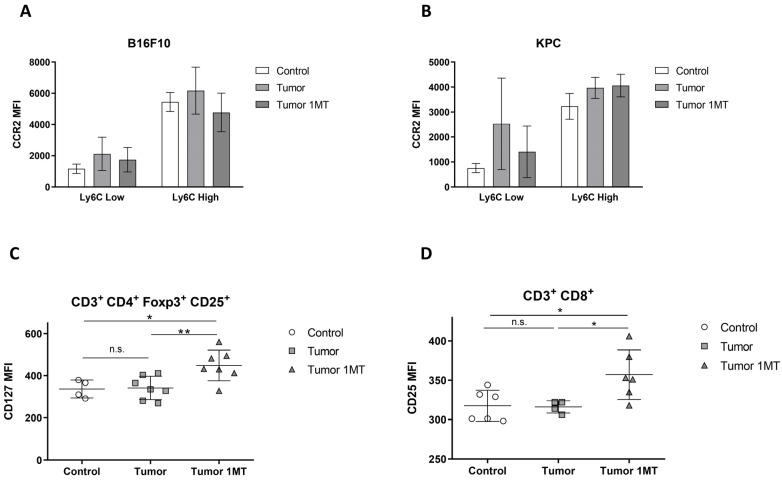
Immune population analysis by flow cytometry on peripheral blood cells after 1-MT treatment in B16F10 and KPC models. CCR2 Mean Fluorescence Intensity (MFI) measured by flow cytometry in B16F10 (**A**) and KPC (**B**) mice. (**C**) CD127 MFI in Tregs population (CD3^+^, CD4^+^, Foxp3^+^, and CD25^+^) in B16F10 model. (**D**) CD25 MFI in CD3^+^ CD8^+^ cells in KPC mice. Significant differences between groups are indicated by * (*p* < 0.05), ** (*p* < 0.005) and n.s. means no significant.

**Figure 7 ijms-24-13005-f007:**
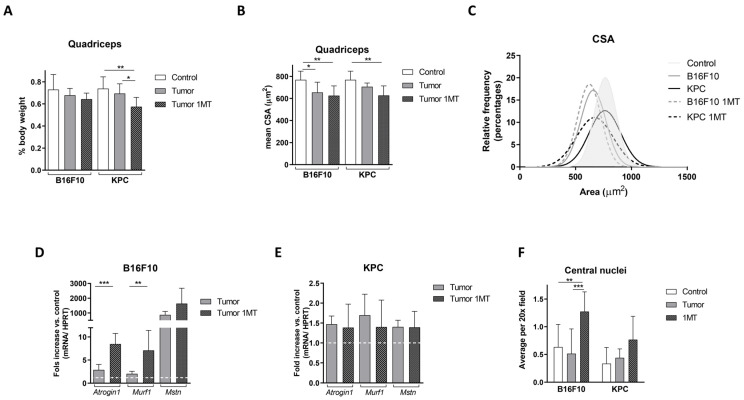
Muscle alterations after 1-MT treatment. (**A**) Quadriceps weight at the time of sacrifice referred to total body weight. Quadriceps cross sectional area measured in H&E sections represented as mean (**B**) and as relative frequency (**C**). *Atrogin1*, *Murf1*, and *Mstn* RNA expression normalized by HPRT, detected by qPCR, in B16F10 (**D**) and KPC (**E**) mice (represented by fold change compared with non-tumor-bearing mice). (**F**) Average number of fibers with central nuclei. Significant differences between groups are indicated by * (*p* < 0.05), ** (*p* < 0.005) and *** (*p* < 0.0005).

**Table 1 ijms-24-13005-t001:** Specific primers used for qPCR.

qPCR Primers	Forward	Reverse
**HPRT**	GGCCAGACTTTGTTGGATTTG	TGCGCTATCTTAGGCTTTGT
**Atrogin1**	AGATTCGCAAGCGTTTGATC	GGGAAAGTGAGACGGAGCAG
**Murf1**	ATTGTAGAAGCCTCCAAGGG	GGTGTTCTTCTTTACCCTCTGTG
**Myostatin**	AGTGGATCTAAATGAGGGCAGT	GTTTCCAGGCGCAGCTTAC

**Table 2 ijms-24-13005-t002:** Antibodies used for flow cytometry analysis.

Monocytes Panel	Manufacturer	Lymphocytes Panel	Manufacturer
anti-CD11b-APC Fire	BioLegend, San Diego, CA, USA	anti-CD4-APC Fire	BioLegend
anti-Ly6C-PerCP	BioLegend	anti-CD3-PE	BioLegend
anti-Ly6G-FITC	BioLegend	anti-CD25-APC	BioLegend
anti-CD43-APC	BioLegend	anti-CD127/IL7Rα BV421	BioLegend
anti-CCR2-PE	BioLegend	anti-CD8-PerCP	BioLegend
anti-CX3CR1-BV421	Miltenyi Biotech, San Diego, CA, USA	anti-Foxp3-FITC	eBioscience, Waltham, MA, USA

## Data Availability

All relevant data is contained within the article or Appendix A. If more information is required, it is available on request from the corresponding authors.

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
