# Peer review of "Tryptophan Modulation in Cancer-Associated Cachexia Mouse Models"

_ijms, 2023, doi:10.3390/ijms241613005_

Round 1

Reviewer 1 Report

General comments

The author’s investigated the role of tryptophan using two cancer associated cachexia syngeneic murine models, melanoma B16F10 and pancreatic adenocarcinoma KPC based. Treatment with 1-methyl-tryptophan, an inhibitor of tryptophan degradation, in both murine models resulted in the restoration of plasmatic tryptophan levels and an improvement in inflammatory signs, lower splenomegaly and oxidative stress, together with enhanced anti-inflammatory features in monocytes and lymphocytes. The manuscript sounds scientific and hold potential for role of Tryptophan modulation in cancer associated. However some points are suggested to improve the overall quality of the manuscript before final publication.

Moderate English editing is required and some typographical errors must be corrected.

Suggestions for authors:

Keywords: Add more relevant keywords.

Abstract: In abstract, rather than general statements, only results should be highlighted to summarize the overall novelty of the manuscript. Rewrite it.

What is control in Figure 1.

Figures are not visible. All figures must be of high resolution. Modify them.

There is no visibility of the statistical analysis. The authors must replace all the figures with high resolution images of high dpi.

Rectify spacing error throughout the manuscript.

Discussion: It should be more precise and informative. It seems very clumsy. Rewrite the section with latest references.

Conclusion: It is missing.

Moderate English editing is required and some typographical errors must be corrected.

Reviewer 2 Report

Please see the attached file. The minor edits could help improve the manuscript.

The quality of the English language is ok. Minor edits required.
